# *Vernonia polysphaera* Baker: Anti-inflammatory activity in vivo and inhibitory effect in LPS-stimulated RAW 264.7 cells

Iara dos Santos da Silva Oliveira[1], Aracélio Viana Colares[2], Flávia de Oliveira Cardoso[3], Carla Junqueira Moragas Tellis[4], Maria do Socorro dos Santos Chagas[4], Maria Dutra Behrens[4], Kátia da Silva Calabrese[3], Fernando Almeida-Souza[3,5]*, Ana Lúcia Abreu-Silva[5]

**1** Rede Nordeste de Biotecnologia, Universidade Federal do Maranhão, São Luís, Maranhão, Brazil, **2** Centro Universitário Dr. Leão Sampaio, Juazeiro do Norte, Ceará, Brazil, **3** Laboratório de Imunomodulação e Protozoologia, Fundação Oswaldo Cruz, Rio de Janeiro, Rio de Janeiro, Brazil, **4** Departamento de Produtos Naturais, Farmanguinhos-Fiocruz, Rio de Janeiro, Rio de Janeiro, Brazil, **5** Pós-graduação em Ciência Animal, Universidade Estadual do Maranhão, São Luís, Maranhão, Brazil

☯ These authors contributed equally to this work.

\* fernandoalsouza@gmail.com

**Data Availability Statement:** All relevant data are within the manuscript and its Supporting Information files.

## Abstract

Species of the *Vernonia* genius are widely distributed across the world. In traditional communities, they are commonly used in popular medicine for the treatment of inflammatory diseases. The objective of the present study was to evaluate the anti-inflammatory activity of *Vernonia polysphaera* Baker hydroalcoholic extract. A λ-carrageenan-induced paw edema and peritonitis model was established in BALB/c mice. The in vitro activity of the extract was measured on LPS-stimulated RAW 264.7 cells. There was no toxic effect on mice or on the cells treated with the extract. Animals treated with *V. polysphaera* extract demonstrated inhibition of paw edema in comparison with the untreated animals at all the analyzed doses. In peritonitis, treatment with the extract at a dose of 500 mg/kg resulted in a lower total leukocyte count in the peritoneal fluid and blood and lower levels of IL-1β, IL-6, TNF-α and PGE-2 than the control group. Cells treated with 50 and 100 μg/mL of the extract exhibited lower levels of nitrite and pro-inflammatory cytokine production and lower COX-2, NF-κB expression. The *V. polysphaera* extract demonstrated an anti-inflammatory effect, interfering with cell migration, reducing pro-inflammatory cytokine levels and COX-2 expression and consequent interference with PGE-2, as well as inhibiting NF-κB transcription.

## Introduction

Inflammation is a physiological process that occurs due to the activation of mechanisms, which cause alterations in the humoral and cellular components. Exposure to a pathogen or tissue injury results in the migration of circulating cells, which are attracted to the inflammatory site by chemotaxis[1]. The regulation of this process involves signals that both initiate and maintain inflammation and those that finalize the process[2].

**Funding:** The present study was funded by the Coordination for the Improvement of Higher Education Personnel (the Coordenação de Aperfeiçoamento de Pessoal de Nível Superior) Brazil (CAPES) [Finance Code 001]; the Research and Scientific and Technological Development Foundation of Maranhão (Fundação de Amparo à Pesquisa e Desenvolvimento Científico e Tecnológico do Maranhão) [grant numbers APP-00844/09, Pronex-241709/2014 to A.L.A.S.]; the National Scientific and Technological Development Council (Conselho Nacional de Desenvolvimento Científico e Tecnológico) [grant numbers 407831/2012.6 and 309885/2017-5 to A.L.A.S.]. Dr. Fernando Almeida-Souza are postdoctoral researcher fellows of CAPES. The funders had no role in study design, data collection and analysis, decision to publish, or preparation of the manuscript.

**Competing interests:** The authors have declared that no competing interests exist.

Following tissue aggression, several immune system components are involved in the inflammatory process. As a consequence of the vasodilatory action of mediators such as prostaglandins, cytokines, tumor necrosis factor alpha (TNF-α) and interferon gama (IFN-ɣ), blood flow at the site intensifies and capillary permeability increases[3]. Due to the latter alteration, retraction of the endothelial cells and adhesion molecule expression occurs in the same cells and leukocytes. Such changes result in the passage of soluble mediators into the vessels and the outflow of cells from the circulation[4]. The mediators include leukotrienes, platelet-activating factor, bradykinins, components of the complement system and cytokines, representing the acute phase of inflammation[5].

Lipopolysaccharide present in the cell wall of gram-negative bacteria can stimulate macrophages and other immune cells to release proinflammatory molecules such as cytokines (IL-1β, IL-6, TNF-α), prostaglandins and nitric oxide (NO). These molecules are known for a variety of biological activities associated with the immunopathology of acute or chronic inflammation, and therefore serve as biomarkers derived from responses generated by a particular pathogenic agent[6]. During the inflammatory process, COX-2 is induced by pro-inflammatory cytokines and growth factors that increase the production of prostaglandins which in turn mediate inflammation, pain and fever[7].

The inflammatory process alters the expression of transcriptional factors, especially in immunologic cells, which regulate inflammation. The transcription factor NF-κB is a crucial component in chronic inflammatory and autoimmune diseases, in which pro-inflammatory cytokines lead to the activation of NF-κB[8].

Modulation of transcription factors such as NF-κB and subsequent pro-inflammatory factors is one of the most effective inflammatory process regulation mechanisms. Several plant-derived secondary metabolites are known to act directly or indirectly on molecules that interfere with the inflammatory mediators[9]. Previous studies of *Vernonia* species have demonstrated the anti-inflammatory properties of their extracts, fractions or constituents[10].

*Vernonia* genus includes species with food, medical, industrial and ornamental uses. *Vernonia polysphaera* Baker plant, popularly known as *assa-peixe* in Brazil, is widely distributed in the Brazilian *cerrado*[11]. In folk medicine its leaves and roots are used in decoction or infusion for diuretic, balsamic, anti-rheumatic purposes, and in cases of flu, bronchitis, pneumonia, and persistent coughs[12]. Most studies with species of *Vernonia* genus in animal models are related to diabetes[13], inflammation[14] and malaria[15].

Due to the applications of species of *Vernonia* genus in folk medicine and scarce content found in the scientific literature about its biological activity, this study aimed to elucidate the molecular mechanism related to the anti-inflammatory activity of *V. polysphaera* Baker in murine model.

## Material and methods

### Plant material

*V. polysphaera* leaves (18.48g) were collected from the Brazilian *cerrado* region in the Araripe National Forest (Ceará) and identified with voucher number 5911 at the Caririense Dárdano de Andrade-Lima Herbarium, Universidade Regional do Cariri. The leaves were dried at 60°C with forced air circulation, ground in a knife mill, macerated and immersed in 70% ethyl alcohol. The supernatant was filtered through filter paper, concentrated at low pressure at 30 to 40°C in a rotary evaporator until the solvent was completely evaporated, packed in an amber bottle and stored at -20°C. For in vitro assays, the extract was solubilized in DMSO (100x concentration) and concentrated DMSO solution was used to prepare the final test concentrations in RPMI 1640 culture medium, with less than 0.5% DMSO in culture medium solution of

extracts. For in vivo assays, extract was solubilized directly in PBS solution. Dilution extracts used in all experiments were prepared immediately before use.

## Direct infusion electrospray ionization mass spectrometry (ESI-MS)

The crude extract of *V. polysphaera* (120mg) was partitioned using increasing polarity solvents such as hexane, dichloromethane and ethyl acetate. These sub-fractions were analyzed for ESI/MS by direct infusion in positive mode to identify the presence of potentially anti-inflammatory compounds. Electrospray ionization mass spectrometric (ESI-MS) measurements were performed in a Bruker Daltonics Amazon SL ion trap. The samples were dissolved in 100% methanol with 0.1% formic acid and introduced into the ESI source through a syringe pump at a flow rate of 100 μL/h. The capillary voltage was 4000 V, and the dry gas flow rate was 5L/min at 250°C. The MS scan was applied for 1.0 minute.

## Animals

Eight week old female BALB/c mice were obtained from the Institute of Science and Technology in Biomodelling (ICTB/FIOCRUZ) and maintained under pathogen-free conditions, at a controlled temperature, with food and water *ad libitum*.

## Ethical statement

The experiments with animals were conducted following the guidelines for experimental procedures of the National Council for the Control of Animal Experimentation (CONCEA) and approved by the Ethics Commission for the Use of Animals of the Fundação Oswaldo Cruz (CEUA-FIOCRUZ), license number LW72/12.

## Acute oral toxicity

The mice were placed in 4 groups (n = 5): treated with PBS (Control), 50, 250 or 500 mg/kg of *V. polysphaera* extract in a final volume of 0.1 mL. Treatment was administered daily by gavage for 14 days. Clinical signs of toxicity, such as piloerection, diarrhea, salivation, convulsions or changes in mobility, respiration rate or muscle tone, as well as mortality were observed during the treatment. On the 15th day, the animals were euthanized with the 250 μL intraperitoneal injection of a 1:1 mixture of ketamine (100mg/mL; Syntec, BRA) and xylazine (20mg/mL; Syntec, BRA). Whole blood was collected by cardiac puncture to obtain serum for the biochemical analysis of creatinine, total bilirubin, uric acid, albumin, urea, alanine aminotransferase (ALT), aspartate aminotransferase (AST) and total protein. Macroscopic evaluation was carried out to check for abnormal findings in the stomach and gut mucosa.

## Paw edema induced by λ-carrageenan

The paw edema was determined by the modified Le Bars et al (2001) method[16]. Mice were separated into six groups of five animals: five groups were pre-treated by gavage with, PBS, 50, 250 or 500 mg/kg of *V. polysphaera*, or dexamethasone (5mg/kg solubilized in PBS, intramuscular route) respectively. After one hour, 30 μL of λ -carrageenan 1% was inoculated into the left hind footpad. A group of animals inoculated with PBS and pre-treated with PBS was maintained as the control group. After 1, 2, 3 and 4 hours of λ-carrageenan inoculation, footpad swelling was measured using a Schnelltaster dial gauge caliper (Kröplin GRBH) and expressed as the difference in thicknesses in millimeters between the inoculated footpad and the non-inoculated footpad. Four hours after λ-carrageenan inoculation, the animals were euthanized and fragments of footpad edema were collected for histological analysis. Tissues were stained

with Giemsa using the modified Wolbach method for the quantification of mast cells. Representative areas in five fields were selected for counting under a light microscope.

## λ -carrageenan-induced peritonitis

The inflammatory peritonitis model was performed according to Guerra et al. (2011)[17]. Animals were divided into six groups and pre-treated as described in item 2.6. After one hour, 250 μL of λ-carrageenan 1% was inoculated intraperitoneally. A group of animals were inoculated with PBS and pre-treated with PBS. After four hours, the animals were euthanized and the peritoneal cavity was harvested with 10 mL of PBS for total and differential cell counting. Blood was collected for total and differential cell counting and sera was obtained to quantify IL-1β, TNF-α, IL-6 and prostaglandin E2 (PGE-2). Total cell counting was carried out in a Neubauer chamber. Differential cell counting was performed from blood or harvest smears slides stained with Panotic.

## Cell culture

RAW 264.7 macrophages, were maintained in RPMI 1640 supplemented with 10% fetal bovine serum (FBS), penicillin (100U/mL) and streptomycin (100μg/mL) at 37°C and 5% $CO_2$ in culture flasks.

## Cytotoxicity assay

Cytotoxicity assay was performed using modified colorimetric MTT [3-(4,5-dimethylthiazol-2-yl)-2,5-diphenyltetrazolium bromide] assay[18]. RAW 264.7 macrophages ($2x10^5$ cells/mL) were incubated for two hours for adhesion in 96 well plates. The medium and non-adherent cells were removed and 100μL of the different extract concentrations (12.5–200 μg/mL) were added. Wells without cells and wells with cells and DMSO 1% were used as blank and control, respectively. After 24 hours, 10 μL of MTT at 5mg/mL was added and incubated for two hours at 37°C and 5% $CO_2$. The plate was then centrifuged at 1.500 rpm for five minutes, the supernatants were removed and the formazan crystals were solubilized with 100 μL of DMSO in a shaker-plate for 15 min. Absorbance was measured in a spectrophotometer at 540 nm wavelength. Cytotoxicity was expressed as a percentage, and the concentration inhibiting 50% of cell growth ($CC_{50}$) was determined using GraphPad Prism 6 software.

## Quantification of endotoxins in *V. polysphaera* extract

Levels of endotoxin present in the *V. polysphaera* extract dilutions (10, 50 and 100 μg/mL, diluted in RPMI medium) were quantified following the recommendations of Pierce™ LAL Chromogenic Endotoxin Quantitation Kit (Thermo Scientific).

## Nitrite, cytokine and PGE-2 quantification in supernatant of RAW 264.7 macrophages stimulated with LPS and treated with *V. polysphaera* extract

RAW 264.7 macrophages ($2x10^6$ cells/mL) in 24-well plates were treated with *V. polysphaera* extract at 10, 50, 100 μg/mL concentrations, or with dexamethasone (5 μg/mL) for one hour, and then stimulated with LPS (10μg/mL). All treatments dilutions were solubilized in RPMI 1640 medium. After 72 hours of treatment, the supernatant was collected for the quantification of nitrite, IL-1β, IL-6, TNF-α and PGE-2. Supernatant nitrite levels were determined by the Griess method[19]. Absorbance was measured in a spectrophotometer at 570 nm. The results were expressed in $NaNO_2$ (μM), based on a standard curve with known concentrations of sodium nitrite (3.1 to 100 μM $NaNO_2$, solubilized in RPMI 1640 medium). Quantification of

IL-1β, TNF-α and IL-6 (BD OptEIA™) and PGE-2 (R&D Systems) was performed following the manufacturer's specifications.

## COX-2 and transcription factor mRNA quantification in RAW 264.7 macrophages stimulated with LPS and treated with *V. polysphaera* extract

Cells were treated with *V. polysphaera* extract and stimulated with LPS as described in item 2.11. The mRNA was extracted with TRIzol reagent (Invitrogen™, Thermo Fischer Scientific, Carlsbad, California) as recommended by the manufacturer. The cDNA synthesis was performed with 1 µg of total RNA using an iScript cDNA Synthesis kit (Bio-Rad, Hercules, CA) following the manufacturer's recommendations. Primers were manufactured by Invitrogen as follows: `5'-GAAGTC TTTGGTCTGGTGCCT-3'/5'-GCTCCTGCTTGAGTATGTCG-3'` for COX-2, `5'-GGA GGCATGTTCGGTAGTGG-3'/5'-CCCTGCGTTGGATTTCGTG-3'` for NF-κB1, `5'-G GCCGGAAGACCTATCCTACT-3'/5'-CTACAGACACAGCGCACACT-3'` for NF-κB2, `5'-AGGCTTCTGGGCCTTATGTG-3'/5'-TGCTTCTCTCGCCAGGAATAC-3'` for RelA, `5'-CCGTACCTGGTCATCACAGAC-3'/5'-CAGTCTCGAAGCTCGATGGC-3'` for RelB, `5'-CTGCTACGTAACACAGTTCCACCC-3'/5'-CATGATGCTTGATCACAT GTCTCG-3'` for B2M. Real Time PCR assays were performed using Power SYBR® Green Master Mix with 5 µL of 1:9 diluted cDNA. A total of 100nM of each primer was used in each reaction at a final volume of 25 µL. The temperature parameters consisted of a hold at 95˚C for 10 min followed by 40 temperature cycles of 95˚C for 15s and 58˚C for 1 min. The relative quantification method ($2^{-\Delta\Delta Ct}$) was applied using RPLP0 (large ribosomal protein P0) or B2M (beta2-microglobulin) mouse gene as an endogenous control. Data were analyzed with the StepOne Software v2.3 package (Applied Biosystems).

## Statistical analysis

The results are represented as mean ± standard deviation and analyzed by Kruskal-Wallis followed by Dunn's multiple comparisons test ($p < 0.05$) or by analysis of variance (two-way ANOVA) followed by Bonferroni's multiple comparisons test ($p < 0.05$).

## Results

### Direct infusion electrospray ionization mass spectrometry (ESI/MS) of *V. polysphaera* extract

The yield of the hydroalcoholic extract was 4.47%, obtained from 18.48g of vegetal material mass. Ions of m/z 301, 317 and 457 corresponding to diosmetin (1), isorhamnetin (2) and oleanolic acid (3) respectively were found in the positive mode ESI spectra in the dichloromethane fraction of crude extract of *V. polysphaera* (Fig 1). The assignment of these ions were based on information from literature[20–22].

### *V. polysphaera* extract did not exhibit acute oral toxicity

No signs of toxicity were observed during treatment with *V. polysphaera* extract, including mortality, clinical signs of toxicity, stomach and gut mucosa appearance. Biochemical parameters of all extract-treated groups presented two-tailed p-values higher than 0.05 when compared to control group treated with PBS (Table 1).

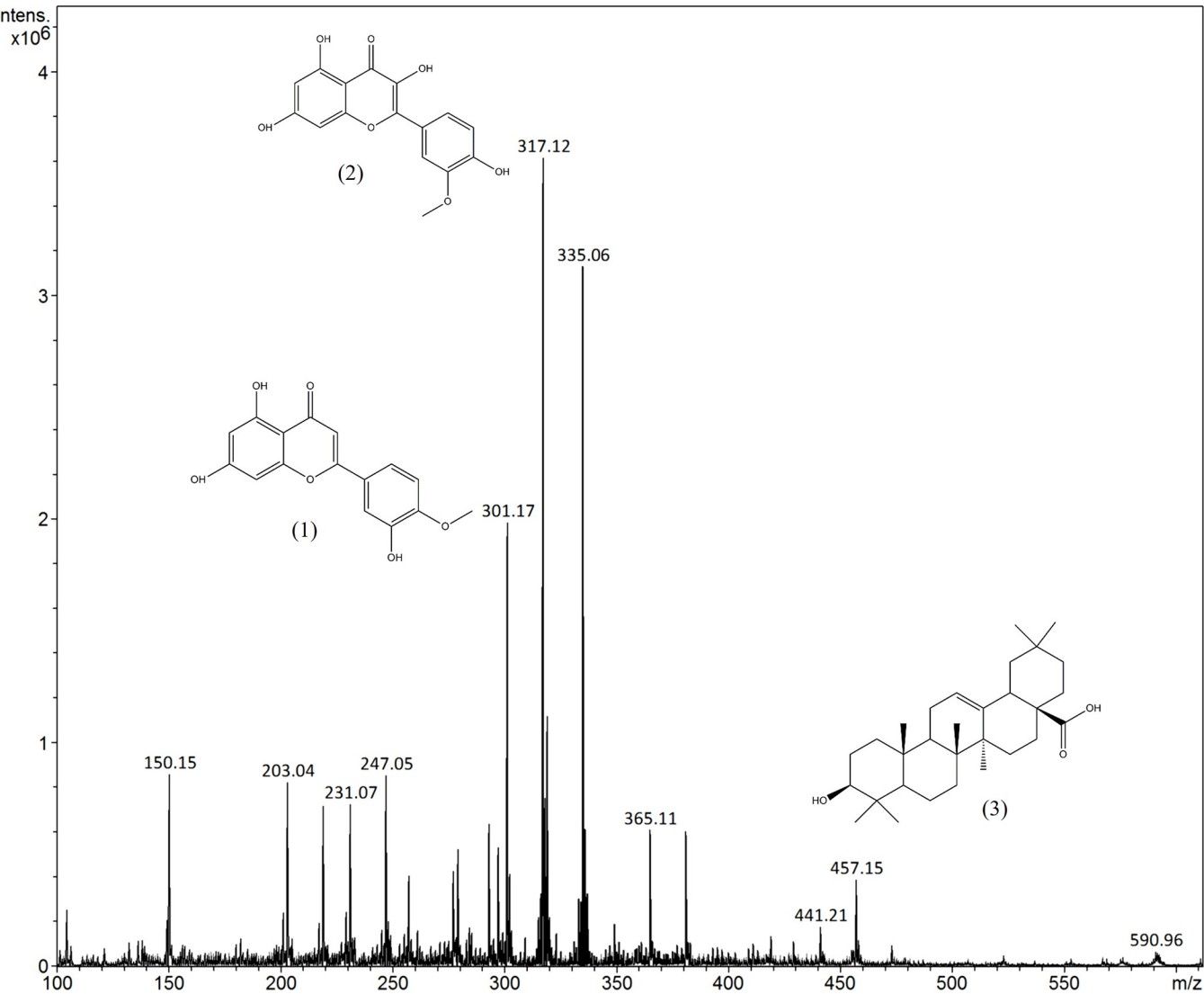

**Fig 1. Off-line ESI-MS of dichloromethane fraction of *Vernonia polysphaera* leaves extract.** The ion at m/z 301 [M+H]$^+$ corresponded to diosmetin (1), the ion at m/z 317 [M+H]$^+$ corresponded to isorhamnetin (2) and the ion at m/z 457 [M+H]$^+$ corresponded to oleanolic acid (3).

### *V. polysphaera* extract inhibited paw edema induced by λ-carrageenan

Groups treated with dexamethasone and *V. polysphaera* extracts demonstrated lower edema thickness than the PBS group three and four hours after treatment (Fig 2A). After four hours, *V. polysphaera* treatment inhibited edema in a dose-dependent manner (Table 2). The groups treated with *V. polysphaera* extract, while they demonstrated lower mean mast cell numbers, did not present statistical differences in comparison with the untreated and stimulated group. A significant lower number of mast cells, p = 0.0014, was only observed in the group treated with dexamethasone (Fig 2B and 2C; S1 Table).

### *V. polysphaera* extract reduced peritonitis induced by λ-carrageenan

In the peritoneal fluid, treatment with *V. polysphaera* extract resulted in a lower number of cells at the 250 and 500 mg/kg dose than in the untreated group, with a similar count to the

**Table 1. Biochemical quantification of liver and kidney parameters in the sera of BALB/c mice treated with *Vernonia polysphaera* hydroalcoholic extract by gavage for 15 days.**

| Parameters | Unit | PBS | *Vernonia polysphaera* (mg/kg/day) | | |
|---|---|---|---|---|---|
| | | | 50 | 250 | 500 |
| Creatinine | mg/dL | 0.20±0.000 | 0.15±0.0577 | 0.15±0.057 | 0.17±0.050 |
| Total bilirubin | mg/dL | 0.67±0.095 | 0.85±0.173 | 0.80±0.115 | 0.87±0.049 |
| Uric acid | mg/dL | 2.32±0.150 | 2.62±0.095 | 2.32±0.377 | 2.47±0.262 |
| Albumin | mg/dL | 4.12±1.577 | 2.85±0.129 | 3.25±1.066 | 3.12±0.537 |
| Urea | mg/dL | 42.27±2.715 | 61.40±1.202 | 60.25±7.747 | 67.57±3.756 |
| ALT | U/dL | 88.50±16.842 | 83.25±11.026 | 93.50±20.371 | 105.66±47.120 |
| AST | U/dL | 157.00±46.357 | 127.50±26.602 | 152.25±38.586 | 203.00±104.043 |
| Total protein | g/dL | 4.05±0.173 | 4.40±0.081 | 4.40±0.163 | 4.30±0.0816 |

ALT: Alanine aminotransferase, AST: Aspartate aminotransferase. Data represent mean ± standard deviation of two experiment performed in quintuplicate.

dexamethasone treated group (Fig 3A). Treatment with 500 mg/kg produced a significantly lower number of leukocytes, p = 0.0013 (Fig 3B). *V. polysphaera* extract at 250 and 500 mg/kg dose resulted in lower levels of IL-1β and IL-6 cytokines than the untreated and stimulated group (Fig 3C and 3D). Lower levels of TNF-α and PGE-2 were only found in the group treated with 500mg/kg of extract (Fig 3E and 3F). The dexamethasone treated group exhibited a reduction in all the inflammatory mediators analyzed in comparison with the untreated and stimulated groups (S2 Table).

### *V. polysphaera* extract reduced levels of pro-inflammatory cytokines and PGE-2 and COX-2 mRNA expression in RAW 264.7 cells stimulated with LPS

There was an absence of cytotoxic activity and endotoxins in the *V. polysphaera* extract at all the concentrations utilized in macrophage treatment. Macrophages stimulated with LPS and treated with *V. polysphaera* extract at 50 and 100 μg/mL exhibited lower amounts of nitrite, IL-1β, IL-6, TNF-α and PGE-2 than the untreated cells. The quantification of nitrites in the group treated with 100 μg/mL of *V. polysphaera* extract was lower even in comparison with the dexamethasone treated group. The downregulation of COX-2 expression was observed only in cells treated with 100 μg/mL of *V. polysphaera* extract (Fig 4, S3 Table).

### *V. polysphaera* extract alters transcription factor mRNA expression in RAW 264.7 cells stimulated with LPS

Treatment with the *V. polysphaera* extract at 100 μg/mL reduced all the transcription factor expression in comparison with the untreated and LPS stimulated cells. Treatment at 50 μg/mL reduced NF-κB2 and RelB expression only, while treatment with 10 μg/mL did not alter the expression of any of the transcription factors analyzed (Fig 5, S4 Table).

## Discussion

The study demonstrated the anti-inflammatory properties of the hydroalcoholic extract of *V. polysphaera* leaves used in Brazil as an alternative medicine for the treatment of inflammatory disorders. The pharmacological activity of certain plants, such as the Asteraceae family, are attributed to the presence of sesquiterpenes[21, 23–25]. Other bioactive compounds with anti-

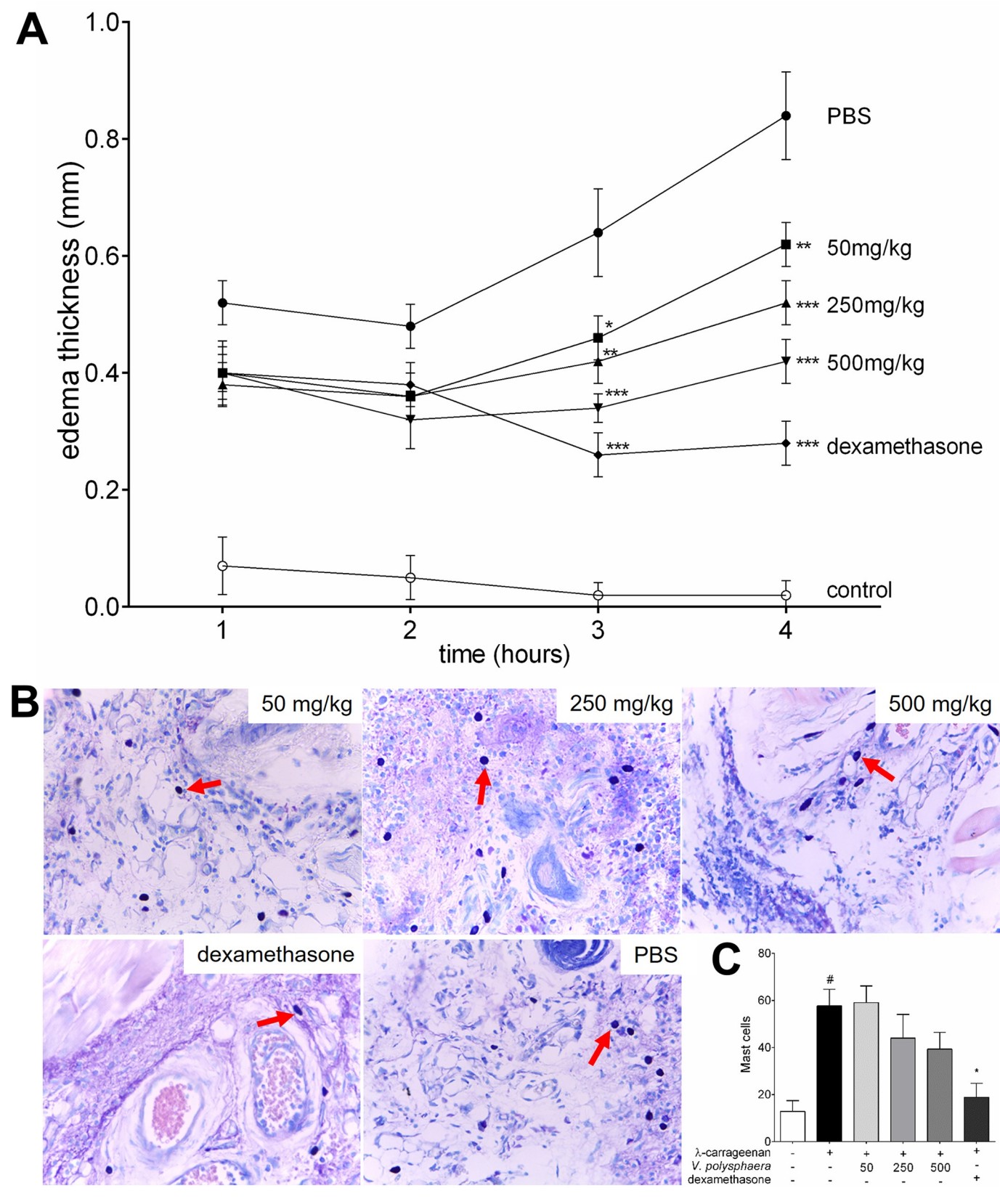

**Fig 2. Paw edema of BALB/ c mice inoculated with of λ-carrageenan and treated with *Vernonia polysphaera* extract.** Edema thickness kinetics (A), histological analysis and quantification of mast cells (B-C) of animals inoculated with 30μL of λ-carrageenan 1%, and treated with 100μL *V. polysphaera* hydroalcoholic extract by gavage or with dexamethasone 5mg/kg via the intramuscular route. Red arrows: mast cells. Data and images are representative of three independent experiments carried out in quintuplicate. #P<0.001 compared with control group; *p<0.05, **P<0.01, ***P<0.001 compared with PBS group, after analysis of variance (two-way ANOVA) followed by Bonferroni's multiple comparisons test for edema kinetics, and Kruskal-Wallis followed by Dunn's multiple comparisons test for mast cell quantification.

inflammatory effects have been isolated from the *Vernonia* genus, such as flavonoids[26], steroid glicoside[27, 28], polysaccharides[29, 30] and chalconoids[31].

The anti-inflammatory activity of three of the compounds characterized by electrospray ionization mass spectrometry (ESI-MS) fingerprinting in the present study was previously described. In addition, they were reported from another *Vernonia* species. Diosmetin, described from *V. patula*[21], was characterized by its anti-inflammatory activity[32]. This compound was found to inhibit the inflammation in the ear of mouse edema and to moderately reduce pro-inflammatory IL-6 or TNF-α levels[33, 34]. The flavonoid isorhamnetin has been described from *V. polyanthes*[20], investigated for its ability to prevent acute inflammation by blocking NF-κB activation[35] and the reduction of oxidative stress[36]. Oleanolic acid, previously reported from *V. auriculifera*[22], is a natural pentacyclic triterpenoid well known for possessing a wide range of biological activities[37, 38]. The anti-inflammatory effects of oleanolic acid have been described by the following mechanisms: the inhibition of elastase, which plays a role in the tissue inflammatory response in rheumatic diseases[39]; the inhibition of complement activity[40]; and the inhibition of cyclooxygenase and lipoxygenase activity[41].

*Vernonia* genus are a source of many chemical compounds. As observed in other plants with potential therapeutic use, the evidence of toxic activity may be a limiting factor for its clinical use. In this way, acute and chronic toxicity studies with medicinal plants should be performed in order to increase safety and use in humans, particularly for the development of pharmaceuticals[42]. While experimental models with rats using the oral route demonstrated the acute toxicity of *Vernonia* spp at concentrations of 300, 2000 and 5000 mg/kg[43–45], no changes were observed in the present study in the biochemical parameters of mice treated with *V. polysphaera* extract at the analyzed doses.

Animals treated with *V. polysphaera* extract demonstrated edema inhibition, suggesting an ascending anti-inflammatory dose-response. Edema occurs due to vascular changes that begin with the momentary constriction of the small vessels and posterior vasodilatation and

**Table 2. Paw edema thickness induced by λ-carrageenan 1% in BALB/c mice treated with *Vernonia polysphaera* hydroalcoholic extract.**

| Group | λ-carrageenan | Dose (mg/kg) | Administration time (% inhibition of edema) | | | |
|---|---|---|---|---|---|---|
| | | | 1 hour | 2 hours | 3 hours | 4 hours |
| Control | - | - | 0.07±0.110 | 0.05±0.084 | 0.02±0.048 | 0.02±0.055 |
| PBS | + | - | 0.52±0.084 | 0.48±0.084 | 0.64±0.167 | 0.84±0.167 |
| *Vernonia polysphaera* extract | + | 50 | 0.40±0.100 | 0.36±0.089 | 0.46±0.084 (28.12)* | 0.62±0.084 (26.19)** |
| | + | 250 | 0.38±0.084 | 0.36±0.089 | 0.42±0.084 (34.37)** | 0.52±0.084 (38.09)*** |
| | + | 500 | 0.40±0.071 | 0.32±0.110 | 0.34±0.055 (46.87)*** | 0.42±0.084 (50.00)*** |
| Dexamethasone | + | 5 | 0.40±0.122 | 0.38±0.084 | 0.26±0.084 (59.37)*** | 0.28±0.048 (66.66)*** |

The data are representative of four independent experiments expressed as mean ± standard deviation of five replicates.

*P<0.05

**p<0.01

***p<0.001 compared to the PBS group after analysis of variance (two-way ANOVA) followed by Bonferroni's multiple comparisons test (p<0.05).

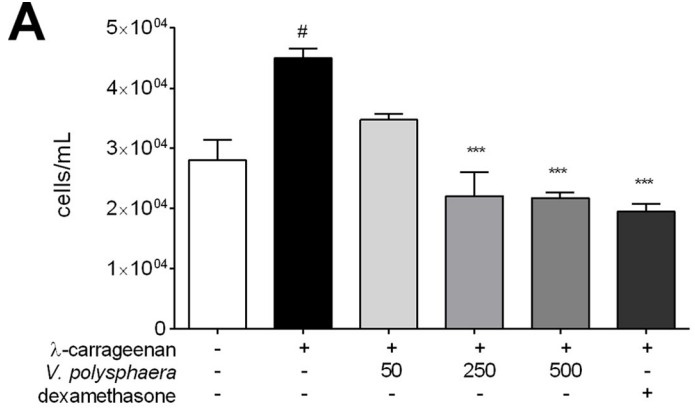

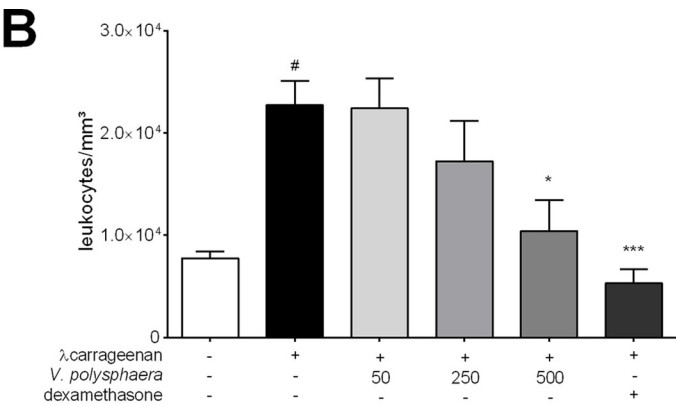

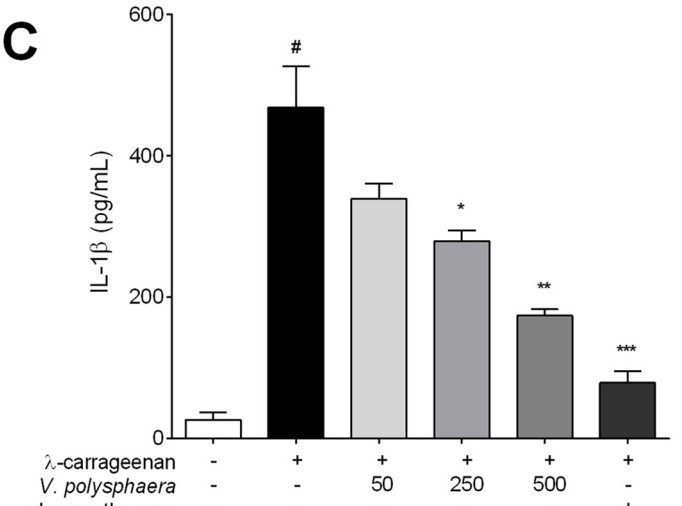

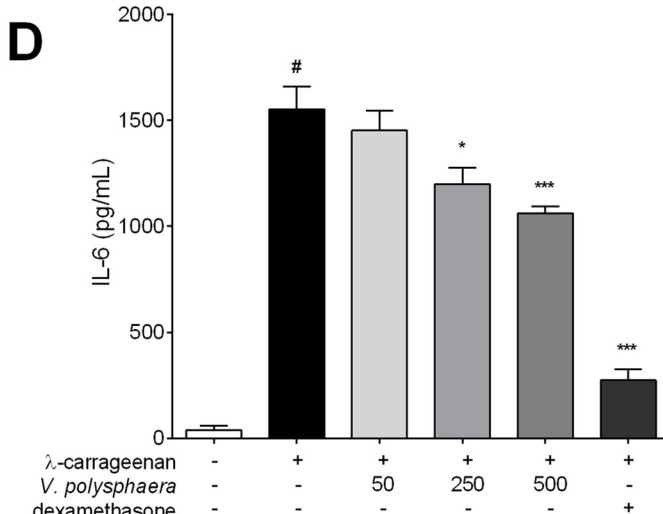

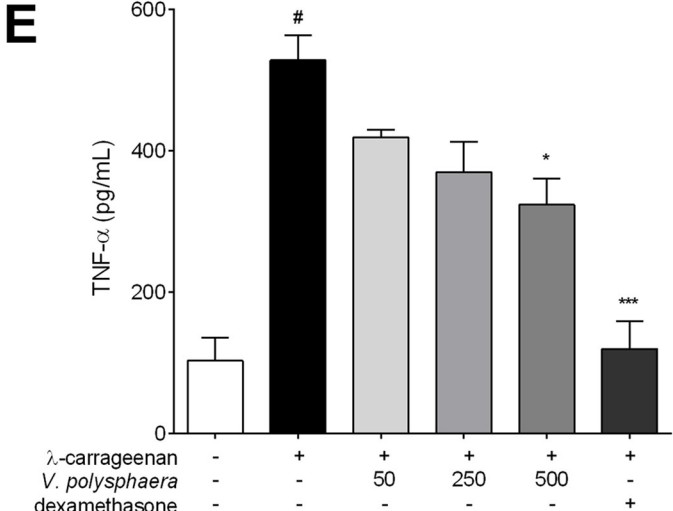

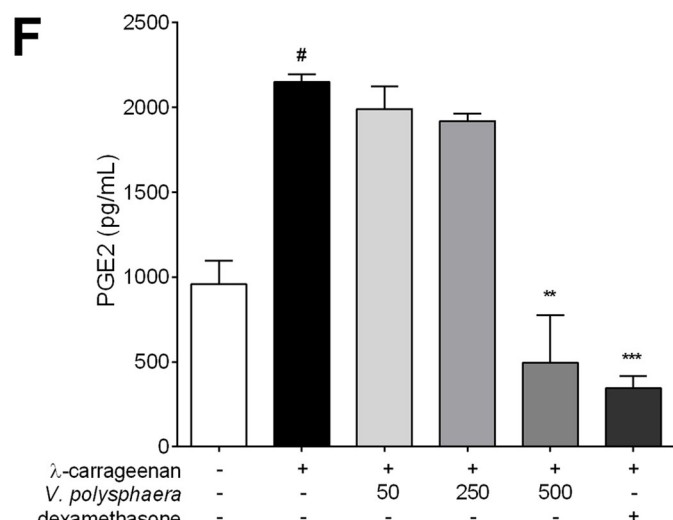

**Fig 3. Peritonitis induced by λ-carrageenan in BALB/ c mice treated with extract of *Vernonia polysphaera*.** Total count of white blood cells (A-B), and cytokine and PGE-2 quantification in sera (C-F) of BALB/c mice after four hours of peritonitis induced by λ-carrageenan 1% treated with *V. polysphaera* hydroalcoholic extract (50, 250, 500mg/kg) or dexamethasone, 5mg/kg. The data are expressed as mean ± standard deviation and are representative of three experiments performed in quintuplicate. #p<0.001 compared with unstimulated group; *p<0.05, **p<0.01, ***p<0.001 compared with the group stimulated with λ-carrageenan only after Kruskal-Wallis analysis followed by Dunn's multiple comparisons test.

leukocyte migration to the trauma area[3]. Leucocytes produce additional inflammatory factors that sustain and potentiate the inflammation. Some compounds prevent cell infiltration by reducing the formation of edema and eicosanoids at the injury site[46], such as *V. polyanthes*, which inhibited ear edema induced by phloglogs[14], and *V. amygdalina*, which reduced leukocyte migration and edema in carrageenan air pockets and paw edema models [47].

Among the cells involved in the pathogenesis of inflammation, mast cells are found mainly in vascularized tissues and display cytoplasmic granules that are rich in a wide variety of mediators[48]. In the present study, the reduction of the mast cells with an increase in the tested concentration of *V. polysphaera* was noteworthy, although the difference was not statistically significant, suggesting that other cells may also be involved in the edema inhibition caused by the treatment.

In carrageenan-induced peritonitis treated with the extract there was a reduction in the cellular migration of the peritoneal fluid and blood, but the differential count did not differ between the treated groups and the stimulated and untreated group. The *V. polysphaera* hydroalcoholic extract therefore has an effect on cell migration, but this action is not cell specific.

Leukocytes require chemotactic elements that facilitate their migration to the inflammation locus in order to neutralize or destroy the aggressor agent. The inhibition of leukocyte migration into the peritoneal cavity can occur via two mechanisms: the inhibition of chemotactic substance production and/or the inhibition of adhesion molecule expression[49].

*V. polysphaera* extract substantially reduced the levels of proinflammatory mediators such as TNF-α, IL-1β, IL-6 and PGE-2 in λ-carrageenan-induced peritonitis when compared to the stimulated and untreated group. Inflammatory mediators such as prostaglandins and proinflammatory cytokines are released into the peritoneal cavity by resident cells and endothelial cells located in the peritoneum, such as macrophages and mast cells[50, 51].

The medicinal use of plants requires validation through in vivo assays and complementary in vitro assays to ascertain the extent of the risks that such compounds may cause to the cells or their interference with inflammation mediators. Cytotoxicity studies with *V. amygdalina* species and cancer cell lines found an $IC_{50}$ around 1.0 μg/mL in LoVo/Adr cells[52], and at 1000 μg/mL in BT-549[53]. For the Vero cell line, an $IC_{50}$ of 60.33 μg/mL was found[54]. While these results demonstrate the high variability of cytotoxicity, probably associated with variability in the chemical composition of *Vernonia* species, in the present study the *V. polysphaera* extract did not exhibit cytotoxicity at the concentrations tested.

RAW 264.7 macrophages provide a good model for assessing anti-inflammatory activity as the inhibition of these pathways leads to the induction and production of pro-inflammatory enzymes and cytokines[55]. This model was therefore used to study in vitro *V polysphaera* activity. Treatment with *V. polysphaera* extract reduced LPS stimulus, inducing low levels of NO, IL-1β, IL-6, TNF-α, and PGE-2, as well as COX-2 and transcription factor mRNA expression in the treated cells.

Macrophages are a major cellular source of COX-2 following exposure to different stimulus. Many cell populations, such as fibroblasts, endothelial cells and monocytes/macrophages can secrete this enzyme in response to different stimulus[56, 57]. Activated macrophages produce

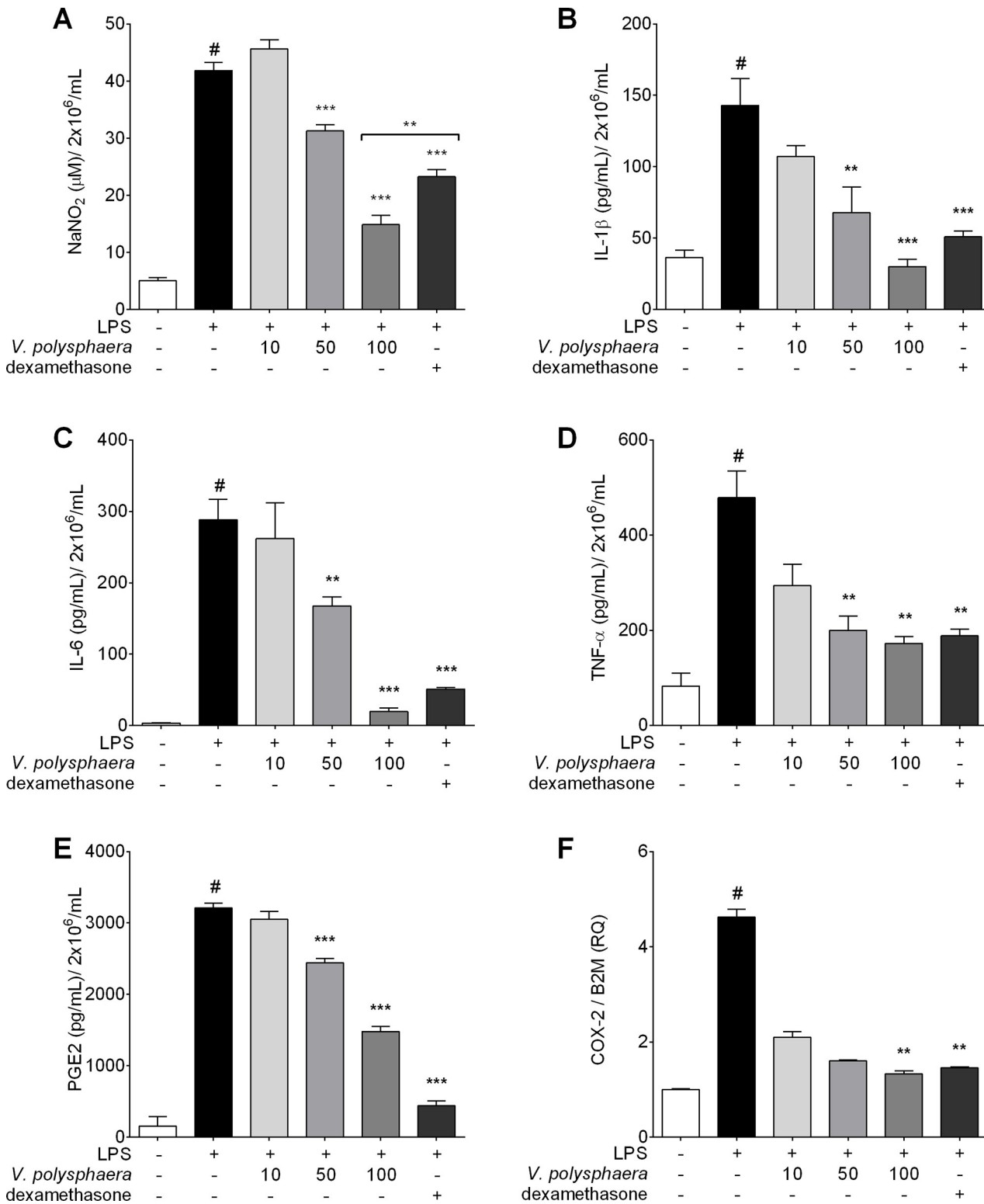

**Fig 4. Proinflammatory factors in RAW 264.7 cells stimulated by LPS and treated with *Vernonia polysphaera* extract.** Levels of nitrite (A), proinflammatory cytokines (B-D), PGE-2 (E) in culture supernatants and mRNA expression of COX-2 (F) in cell culture treated with *V. polysphaera* (10, 50 or 100μg/mL) or dexamethasone, 100 μM. Data are representative of two independent experiments performed at least in quadruplicate. #P <0.001 compared with the group without stimulation or treatment; **p<0.01, ***p<0.001 compared with the stimulated and untreated group after Kruskal-Wallis analysis followed by Dunn's multiple comparisons test.

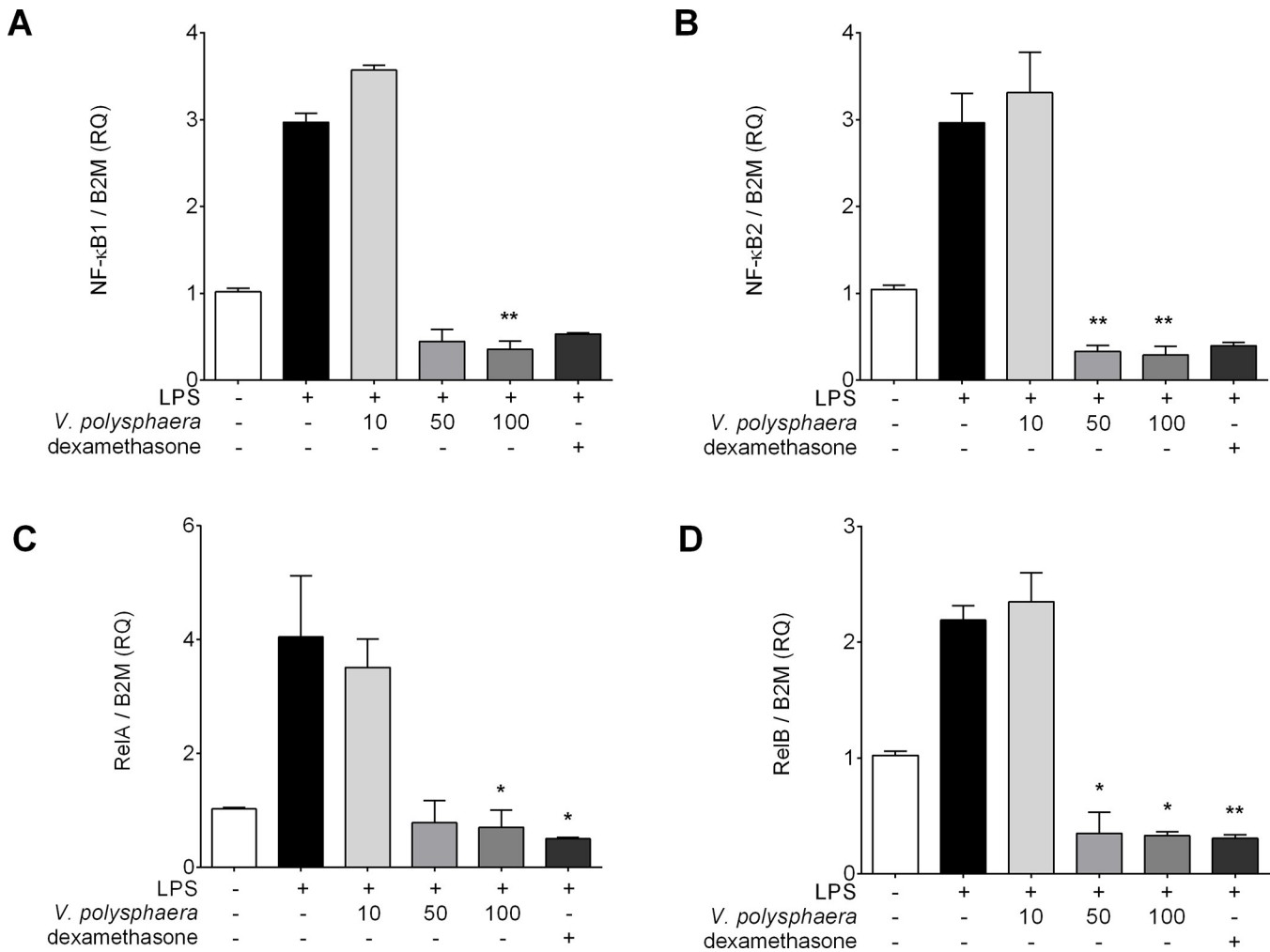

**Fig 5. Proinflammatory transcriptional factor expression in RAW 264.7 cells stimulated by LPS and treated with *Vernonia polysphaera* hydroalcoholic extract.** NF-κB 1 (A), NF-κB 2 (B), RelA (C) and RelB (D) expression in cell culture treated with *V. polysphaera* (10, 50 or 100μg/mL) or dexamethasone, 100 μM. Data are representative of two independent experiments performed at least in quadruplicate. #p <0.001 compared with the group without stimulation and without treatment. P<0.05, **p<0.01 compared with the group stimulated with untreated LPS after Kruskal-Wallis analysis followed by Dunn's multiple comparisons test.

PGE-2 by the proinflammatory induction of the COX-2 enzyme[58]. The present study demonstrated that *V. polysphaera* extract suppressed COX-2 expression, which subsequently suppressed PGE-2 expression in cells stimulated by LPS.

NF-κB family proteins are an important cellular transcription factor which regulates the expression of the COX-2 inflammatory enzyme and the pro-inflammatory cytokines. NF-κB exist in the cytoplasm of most cells in an inert form, without binding to the DNA. Its activation predominantly occurs by phosphorylation and degradation of the IκB proteins[59]. Abnormal activation of NF-κB is an important factor in the induction of inflammation, cancer and other pathologies[60]. The present study demonstrated that *V. polysphaera* extract was able to reduce the expression levels of the NF-κB transcription factors analyzed.

Studies of other *Vernonia* species and molecules isolated from the same also addressed their effects on NF-κB pathway regulation. Compounds extracted from the leaves of *V. amygdalina* were evaluated for the immunomodulation of a series of transcription factors (NF-κB, STAT3

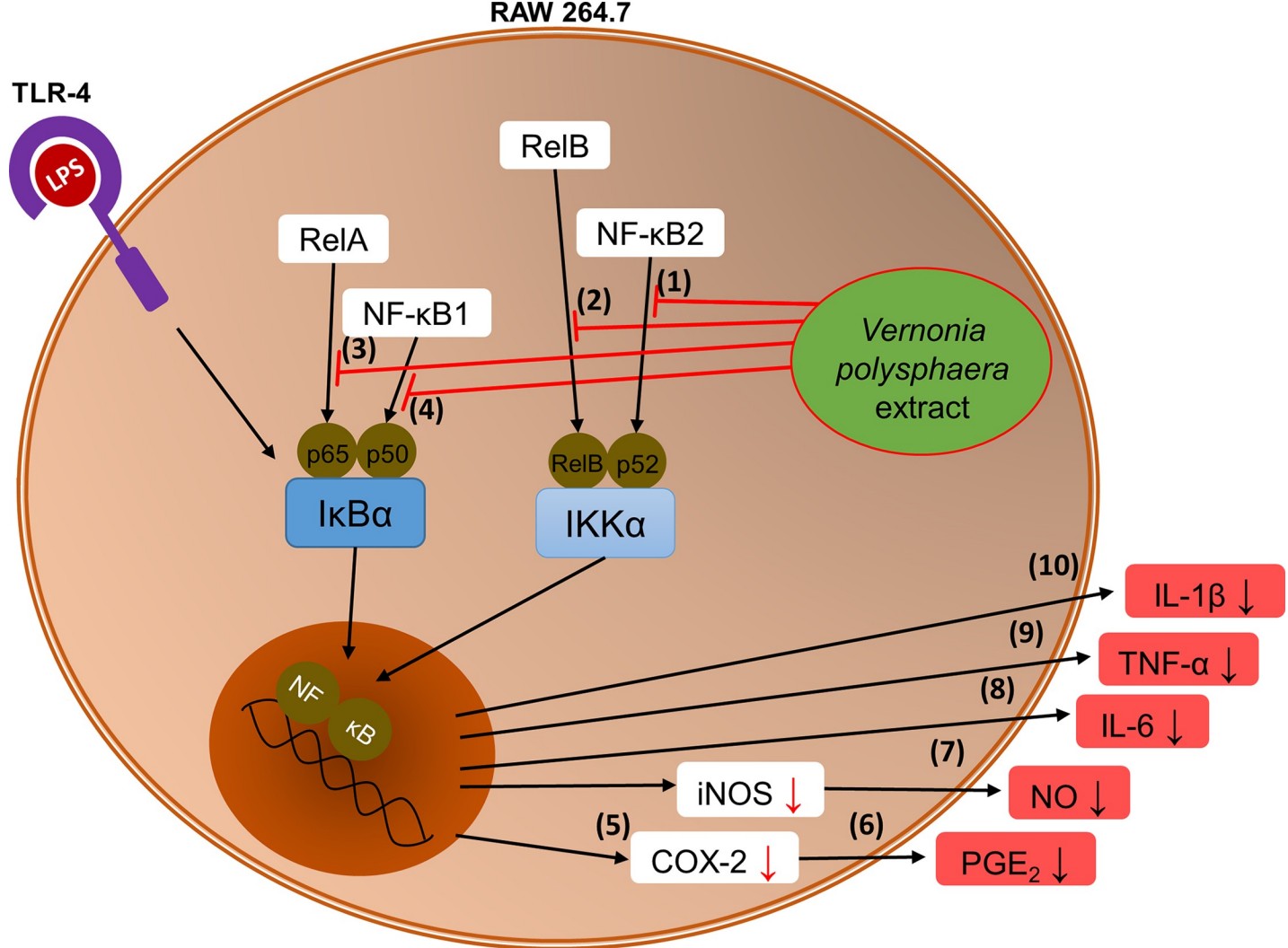

**Fig 6. *Vernonia polysphaera* extract activity in RAW 264.7 cells stimulated with LPS.** Extract inhibition of NF-κB transcription factors (1–4) preventing DNA-NF-κB binding, and thus suppressing the transcription of COX-2 (5), and consequent PGE-2 production (6), as well as inducing low levels of inflammatory mediators such as NO (7), TNF-α (8), IL-6 (9), IL-1β (10).

and Nrf2) involved in maintaining the chronic inflammatory condition of human degenerative diseases. Vernolide, a recently discovered molecule from plants of the *Vernonia* genus, emerged as a potent inhibitor of NF-κB[61]. Compounds of *V. cinerea* flower extract were isolated to assess their ability to inhibit nitric oxide (NO) production and NF-κB activity in cancer. The compounds inhibited TNF-α-induced NF-κB activity[25].

Overall, the results complement each other, with the *V. polysphaera* extract inhibition of NF-κB transcription factor preventing DNA-NF-κB binding, and thus suppressing the transcription of the IL-1β, IL-6, TNF-α and COX-2 genes, evidenced by low levels of these inflammatory mediators and consequent interference in PGE-2 (Fig 6). *V. polysphaera* extract has an anti-inflammatory effect, as it is able to decrease edema formation and cellular migration, reducing pro-inflammatory cytokines levels and COX-2 expression with consequent interference in PGE-2 production, as well as interfering with inflammatory cellular transcription factors.

## Supporting information

**S1 Table. Mast cells quantification.**
(DOCX)

**S2 Table. Peritonitis induced by λ-carrageenan in BALB/ c mice treated with extract of *Vernonia polysphaera*.**
(DOCX)

**S3 Table. Proinflammatory factors in RAW 264.7 cells stimulated by LPS and treated with *Vernonia polysphaera* extract.**
(DOCX)

**S4 Table. Proinflammatory transcriptional factor expression in RAW 264.7 cells stimulated by LPS and treated with *Vernonia polysphaera* hydroalcoholic extract.**
(DOCX)

## Acknowledgments

The authors would like to thank the Program for Technological Development in Health Tools-PDTIS-FIOCRUZ for the use of its facilities. The present study was funded by the Coordination for the Improvement of Higher Education Personnel (the Coordenação de Aperfeiçoamento de Pessoal de Nível Superior) Brazil (CAPES) [Finance Code 001]; the Research and Scientific and Technological Development Foundation of Maranhão (Fundação de Amparo à Pesquisa e Desenvolvimento Científico e Tecnológico do Maranhão) [grant numbers APP-00844/09, Pronex-241709/2014 to A.L.A.S]; the National Scientific and Technological Development Council (Conselho Nacional de Desenvolvimento Científico e Tecnológico) [grant numbers 407831/2012.6 and 309885/2017-5 to A.L.A.S.]. Dr. Fernando Almeida-Souza are postdoctoral researcher fellows of CAPES. The funders had no role in study design, data collection and analysis, decision to publish, or preparation of the manuscript.

## Author Contributions

**Conceptualization:** Iara dos Santos da Silva Oliveira, Carla Junqueira Moragas Tellis, Maria Dutra Behrens, Kátia da Silva Calabrese, Fernando Almeida-Souza, Ana Lúcia Abreu-Silva.

**Formal analysis:** Carla Junqueira Moragas Tellis, Maria Dutra Behrens, Kátia da Silva Calabrese, Fernando Almeida-Souza, Ana Lúcia Abreu-Silva.

**Funding acquisition:** Maria Dutra Behrens, Kátia da Silva Calabrese, Fernando Almeida-Souza, Ana Lúcia Abreu-Silva.

**Methodology:** Iara dos Santos da Silva Oliveira, Aracélio Viana Colares, Flávia de Oliveira Cardoso, Carla Junqueira Moragas Tellis, Maria do Socorro dos Santos Chagas, Fernando Almeida-Souza.

**Resources:** Maria Dutra Behrens, Kátia da Silva Calabrese, Ana Lúcia Abreu-Silva.

**Supervision:** Fernando Almeida-Souza, Ana Lúcia Abreu-Silva.

**Writing – original draft:** Iara dos Santos da Silva Oliveira, Carla Junqueira Moragas Tellis, Fernando Almeida-Souza.

**Writing – review & editing:** Iara dos Santos da Silva Oliveira, Aracélio Viana Colares, Flávia de Oliveira Cardoso, Maria do Socorro dos Santos Chagas, Maria Dutra Behrens, Kátia da Silva Calabrese, Fernando Almeida-Souza, Ana Lúcia Abreu-Silva.

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
