## [Decision Letter · Decision Letter 0]

11 Oct 2019

PONE-D-19-24305

Vernonia polysphaera Baker: anti-inflammatory activity in vivo and inhibitory effect in LPS-stimulated RAW 264.7 cells

PLOS ONE

Dear Dr. Almeida-Souza,

Thank you for submitting your manuscript to PLOS ONE. After careful consideration, we feel that it has merit but does not fully meet PLOS ONE’s publication criteria as it currently stands. Therefore, we invite you to submit a revised version of the manuscript that addresses the points raised during the review process.

Your manuscript was reviewed by two experts and  we received positive feedback from both of them. Please adresss technical issues raised by the reviewers. 

We would appreciate receiving your revised manuscript by Nov 24 2019 11:59PM. To enhance the reproducibility of your results, we recommend that if applicable you deposit your laboratory protocols in protocols.io, where a protocol can be assigned its own identifier (DOI) such that it can be cited independently in the future. For instructions see: http://journals.plos.org/plosone/s/submission-guidelines#loc-laboratory-protocols

We look forward to receiving your revised manuscript.

Kind regards,

Partha Mukhopadhyay, Ph.D.

Academic Editor

PLOS ONE

Journal Requirements:

1. Thank you for including your funding statement; "The funders had no role in study design, data collection and analysis, decision to publish, or preparation of the manuscript."

Please provide an amended Funding Statement that declares *all* the funding or sources of support received during this specific study (whether external or internal to your organization) as detailed online in our guide for authors at http://journals.plos.org/plosone/s/submit-now.  

Please state what role the funders took in the study.  If any authors received a salary from any of your funders, please state which authors and which funder. If the funders had no role, please state: "The funders had no role in study design, data collection and analysis, decision to publish, or preparation of the manuscript."

Reviewers' comments:

Reviewer's Responses to Questions

**Comments to the Author**

1. Is the manuscript technically sound, and do the data support the conclusions?

Reviewer #1: Yes

Reviewer #2: Yes

2. Has the statistical analysis been performed appropriately and rigorously? 

Reviewer #1: Yes

Reviewer #2: Yes

3. Have the authors made all data underlying the findings in their manuscript fully available?

Reviewer #1: Yes

Reviewer #2: Yes

4. Is the manuscript presented in an intelligible fashion and written in standard English?

Reviewer #1: Yes

Reviewer #2: Yes

5. Review Comments to the Author

Reviewer #1: In the present study, the authors evaluated the anti-inflammatory effects of hydroalcoholic extract of a plant Vernonia polysphaera Baker in mouse peripheral and central inflammatory lesion models. The authors performed comprehensive analysis in both in vivo and in vitro effects of said plant extract. The experiments are carried out with balanced designs and results are properly interpreted. There are a few minor details that requires addressing:

Methods: Please indicate type of culture medium used to reconstitute the extract for in vitro and in vivo use (Line 101). The extract was prepared in 3 different ways: in DMSO, in culture medium and/or in PBS. For culture medium and PBS prepared extract, which was used in what experiments? I suggest including the info in respective paragraphs of experiments. Same goes when mentioning control or standard reagents for measurements, e.g. nitrite standards prepared in culture medium if that is the case.

Table 1: include two-tailed p-values for each extract-treated group to control group.

Reviewer #2: In this manuscript, the authors described the action by which organic extracts from Vernonia polysphaera Baker reduces inflammation both in vivo and in vitro. The study was well presented and performed with care. I have the following minor points, mostly relating to the model presented in figure 6.

1) Does Vernonia polysphaera extracts reduce both NFkB activation and pro-inflammatory cytokine production by independent actions as shown in Figure 6? Or is the reduction of pro-inflammatory cytokine a result of reduced NFkB activity?

2) Does Vernonia polysphaera extracts inhibit NFkB signal by interfering with the processing of NFkB precursor or its generation? Ideally, a simple set of experiments that should be included is western blotting for both the precursor and processed NFkB subunits in Vernonia polysphaera treated vs. non-treated groups.

I am a traditionalist concerning quantifying signaling molecules using PCR method and would like to see measurable protein changes. I leave it up to the editor to decide whether this condition needs to be met.

6. PLOS authors have the option to publish the peer review history of their article (what does this mean?). If published, this will include your full peer review and any attached files.

Reviewer #1: No

Reviewer #2: No

---

## [Author Response · Author response to Decision Letter 0]

17 Oct 2019

As requested in revision, follow below Acknowledgments item with amended funding statement:

The authors would like to thank the Program for Technological Development in Health Tools-PDTIS-FIOCRUZ for the use of its facilities. The present study was funded by the Coordination for the Improvement of Higher Education Personnel (the Coordenação de Aperfeiçoamento de Pessoal de Nível Superior) Brazil (CAPES) [Finance Code 001]; the Research and Scientific and Technological Development Foundation of Maranhão (Fundação de Amparo à Pesquisa e Desenvolvimento Científico e Tecnológico do Maranhão) [grant numbers APP-00844/09, Pronex-241709/2014 to A.L.A.S]; the National Scientific and Technological Development Council (Conselho Nacional de Desenvolvimento Científico e Tecnológico) [grant numbers 407831/2012.6 and 309885/2017-5 to A.L.A.S.]. Dr. Fernando Almeida-Souza are postdoctoral researcher fellows of CAPES. The funders had no role in study design, data collection and analysis, decision to publish, or preparation of the manuscript.

---

## [Decision Letter · Decision Letter 1]

1 Nov 2019

Vernonia polysphaera Baker: anti-inflammatory activity in vivo and inhibitory effect in LPS-stimulated RAW 264.7 cells

PONE-D-19-24305R1

Dear Dr. Almeida-Souza,

We are pleased to inform you that your manuscript has been judged scientifically suitable for publication and will be formally accepted for publication once it complies with all outstanding technical requirements.

With kind regards,

Partha Mukhopadhyay, Ph.D.

Section Editor

PLOS ONE

Additional Editor Comments (optional):

Reviewers' comments:

Reviewer's Responses to Questions

**Comments to the Author**

1. If the authors have adequately addressed your comments raised in a previous round of review and you feel that this manuscript is now acceptable for publication, you may indicate that here to bypass the “Comments to the Author” section, enter your conflict of interest statement in the “Confidential to Editor” section, and submit your "Accept" recommendation.

Reviewer #1: All comments have been addressed

Reviewer #2: All comments have been addressed

2. Is the manuscript technically sound, and do the data support the conclusions?

Reviewer #1: Yes

Reviewer #2: Yes

3. Has the statistical analysis been performed appropriately and rigorously? 

Reviewer #1: Yes

Reviewer #2: Yes

4. Have the authors made all data underlying the findings in their manuscript fully available?

Reviewer #1: Yes

Reviewer #2: Yes

5. Is the manuscript presented in an intelligible fashion and written in standard English?

Reviewer #1: Yes

Reviewer #2: Yes

6. Review Comments to the Author

Reviewer #1: After reading through the revised manuscript, the previous comments are all addressed.

I hope the authors would describe their revisions in the manuscript in response letter as well, as changes in text especially in large paragraphs could result in shifting line numbers and difficulties in tracing back those changes.

Reviewer #2: The authors have addressed all concerns in the revised manuscript and I have no further questions.

7. PLOS authors have the option to publish the peer review history of their article (what does this mean?). If published, this will include your full peer review and any attached files.

Reviewer #1: No

Reviewer #2: No

---

## [Editor Report · Acceptance letter]

5 Dec 2019

PONE-D-19-24305R1 

*Vernonia polysphaera* Baker: anti-inflammatory activity in vivo and inhibitory effect in LPS-stimulated RAW 264.7 cells 

Dear Dr. Almeida-Souza:

I am pleased to inform you that your manuscript has been deemed suitable for publication in PLOS ONE. Congratulations! Your manuscript is now with our production department. 

With kind regards,

on behalf of

Dr. Partha Mukhopadhyay 

Section Editor

PLOS ONE